# Functional Progression after Dose Suspension or Discontinuation of Nintedanib in Idiopathic Pulmonary Fibrosis: A Real-Life Multicentre Study

**DOI:** 10.3390/ph17010119

**Published:** 2024-01-17

**Authors:** Barbara Ruaro, Andrea Salotti, Nicolò Reccardini, Stefano Kette, Beatrice Da Re, Salvatore Nicolosi, Umberto Zuccon, Marco Confalonieri, Lucrezia Mondini, Riccardo Pozzan, Michael Hughes, Paola Confalonieri, Francesco Salton

**Affiliations:** 1Department of Pulmonology, University Hospital of Cattinara, 34149 Trieste, Italy; salottiandrea@gmail.com (A.S.); nico.recca@gmail.com (N.R.); stefano.kette@libero.it (S.K.); beatricedare95@gmail.com (B.D.R.); salvatore.nicolosi@studenti.units.it (S.N.); lmondinifr@gmail.com (L.M.); paola.confalonieri.24@gmail.com (P.C.);; 2Pulmonology Unit, General Hospital “Santa Maria degli Angeli”, 33170 Pordenone, Italy; umberto.zuccon@aas5.sanita.fvg.it (U.Z.); riccardo.pozzan@asugi.sanita.fvg.it (R.P.); 3Division of Musculoskeletal and Dermatological Sciences, Faculty of Biology, Medicine and Health, The University of Manchester Salford Royal NHS Foundation Trust, Manchester M6 8HD, UK; michael.hughes-6@manchester.ac.uk

**Keywords:** idiopathic pulmonary fibrosis (IPF), interstitial pneumonia, interstitial lung disease (ILD), Nintedanib (NTD), pulmonary function tests (PFTs), high-resolution computed tomography (HRTC)

## Abstract

Background. Idiopathic pulmonary fibrosis (IPF) is a chronic interstitial lung disease with rapidly progressive evolution and an unfavorable outcome. Nintedanib (NTD) is an antifibrotic drug that has been shown to be effective in slowing down the progression of the disease. The aim of our study was to examine the efficacy, especially in terms of the functional decline, and the safety profile of NTD in patients treated with the recommended dose and subjects who reduced or suspended the therapy due to the occurrence of adverse reactions. Methods. We conducted a real-life retrospective study based on the experience of NTD use in two centers between 2015 and 2022. Clinical data were evaluated at baseline, at 6 and 12 months after the NTD introduction in the whole population and in subgroups of patients who continued the full-dose treatment, at a reduced dosage, and at the discontinuation of treatment. The following data were recorded: the demographic features, IPF clinical features, NTD therapeutic dosage, tolerability and adverse events, pulmonary function tests (PFTs), the duration of treatment upon discontinuation, and the causes of interruption. Results. There were 54 IPF patients who were included (29.6% females, with a median (IQR) age at baseline of 75 (69.0–79.0) years). Twelve months after the introduction of the NTD therapy, 20 (37%) patients were still taking the full dose, 11 (20.4%) had reduced it to 200 mg daily, and 15 (27.8%) had stopped treatment. Gastrointestinal intolerance predominantly led to the dose reduction (13.0%) and treatment cessation (20.4%). There were two deaths within the initial 6 months (3.7%) and seven (13.0%) within 12 months. Compared to the baseline, the results of the PFTs remained stable at 6 and 12 months for the entire NTD-treated population, except for a significant decline in the DLCO (% predicted value) at both 6 (38.0 ± 17.8 vs. 43.0 ± 26.0; *p* = 0.041) and 12 months (41.5 ± 15.3 vs. 44.0 ± 26.8; *p* = 0.048). The patients who continued treatment at the full dose or a reduced dosage showed no significant differences in the FVC and the DLCO at 12 months. Conversely, those discontinuing the NTD exhibited a statistically significant decline in the FVC (% predicted value) at 12 months compared to the baseline (55.0 ± 13.5 vs. 70.0 ± 23.0; *p* = 0.035). Conclusions. This study highlights the functional decline of the FVC at 12 months after the NTD initiation among patients discontinuing therapy but not among those reducing their dosage.

## 1. Introduction

Idiopathic pulmonary fibrosis (IPF) is a rare chronic parenchymal lung disease, although it is the most common among idiopathic interstitial pneumonias with an estimated incidence of 3–9 cases per 100,000 individuals in Europe and North America [1,2,3,4]. Furthermore, IPF is a debilitating life-limiting condition with a median survival time ranging from 2.5 to 3.5 years [1,2,3,4]. As the word “idiopathic” suggests, IPF is a spontaneously occurring, specific form of irreversible fibrosing interstitial pneumonia, whose etiology is unknown. The physio-pathological process of the chronic abnormal deposition of extracellular matrix is limited to the lung, and it is usually associated with a characteristic usual interstitial pneumonia (UIP) pattern on a high-resolution computed tomography (HRCT) and a lung histology [5,6,7,8,9]. The same pattern can also be found in non-idiopathic interstitial lung disease (ILD). Therefore, the term UIP is used to refer to radiographic and histologic patterns and is not a synonym of IPF, which connotes the disease [8,9,10,11,12,13,14]. IPF has a heterogeneous clinical manifestation with relatively nonspecific and insidious signs and symptoms, such as dyspnea on exertion, a non-productive cough, and inspiratory crackles on lung examination [12,13,14,15,16]. Due to this blurred presentation, IPF can be easily confused with other respiratory diseases, and a multidisciplinary assessment by the pulmonologist, the radiologist, and the pathologist can be necessary to establish the diagnosis. These issues frequently cause the delay of a diagnosis, which can be very challenging [9,10,11,12,13,14]. The symptoms associated with IPF cause a progressive reduction in physical activity and muscle endurance, often leading to a poor quality of life due to consequent social isolation and mood disorders such as anxiety and depression [8,9,10,11,12,13,14,15]. IPF also has a considerable variability in its pace of progression, being rapidly progressive or having a slow steady decline over years [9,10,11,12,13,14,15,16]. From the time of the diagnosis, the estimated survival time varies between 3 and 5 years [3,7]. In people suffering from IPF, pulmonary function tests (PFTs) are often altered. Due to an aberrant lung repair process leading to tissue scar formation and lung stiffness, these patients show alveolar structure abnormalities, and consequently gas exchange can be significantly impaired. A common finding in PFTs is reduced lung function (the forced vital capacity (FVC) and the carbon monoxide diffusing capacity (DLCO) [8,9,10,11]. Among the most robust predictors of mortality, there are changes in the pulmonary function tests over time on which some studies are focused [11,12,13,14,15,16]. A recent prospective study found that the lung function decline is constant over a prolonged period of time, supporting the progressive nature of the disease [12,13,14,15,16]. Current treatment strategies advocate a multidisciplinary approach to diagnosis and management with a holistic approach encompassing pulmonary rehabilitation, oxygen therapy, palliation, and anti-fibrotic therapies [2,3]. Among the available therapeutic strategies, there are antifibrotic therapies, such as pirfenidone and nintedanib, which can only reduce respiratory function decline, improving survival but not stopping or reversing the disease progression [9,14,15,16,17,18,19,20]. Nintedanib (NTD) is an intracellular inhibitor of various non-receptor tyrosine kinases (nRTKs) and receptor tyrosine kinases (RTKs). Its targets include fibroblast growth factor receptor (FGFR1, FGFR2, FGFR3), vascular endothelial growth factor (VEGFR1, VEGFR2, VEGFR3), platelet-derived growth factor (PDGFRα and PDGFRβ), colony-stimulating factor receptor 1 (CSF1R), and FMS-like tyrosine kinase three (FLT3) [21,22,23,24,25,26,27]. All the latter are critical players in fibroblast proliferation, migration, and differentiation [9,16,22,23,24,25,26,27,28,29]. Currently, there is limited evidence regarding the evolution of the disease and the trend of a decline in lung function among patients who underwent treatment with NTD but had their dosage reduced or discontinued due to adverse effects [9,16,25,26,27,28,29,30,31,32]. The aim of our study was to investigate the functional decline based on therapeutic dosing or a discontinuation of therapy and to further examine the efficacy and safety profile of NTD.

## 2. Results

We included 54 patients with a confirmed diagnosis of IPF in our two-center retrospective study, of whom 38 (70.4%) were male, with a median (IQR) age at the baseline of 75.0 (69.0–79.0) years and a median (IQR) age of onset of 73.5 (68.0–77.0) years. GERD was the most prevalent comorbidity (16 (29.6%)), followed by cancer (7 (13.0%)), and COPD (5 (9.3%)). Thirty-one (57.4%) patients were ever-smokers. The most common concomitant medications were PPI (35 (64.8%)) and OCS (17 (31.5%)), and only a minority of patients were on immunosuppressants. Radiologically, 38 (70.4%) subjects presented a UIP pattern, while 9 (16.7%) presented a NSIP pattern at the HRCT. All cases considered were discussed in multidisciplinary meetings, and 10 cases (18%), in which the differential diagnosis was not certain, underwent lung biopsy, which confirmed the diagnosis of IPF. The baseline characteristics, collected at the time of the first visit in our center and before any treatments’ modification, are reported in Table 1.

All included patients started therapy at baseline with a dosage of 300 mg per day; 12 months later, 20 (37.0%) were still taking Nintedanib at a full dose, 11 (20.4%) had reduced the dosage to 200 mg daily, and 15 (27.8%) had stopped treatment. There were six (11.1%) subjects who did not report any data; of these, some patients were not considered as the functional follow-up did not take place as we scheduled, so it was decided not to include them. Some of the patients, coming from centers far from ours, preferred to continue treatment near their home. Additionally, seven (13.0%) patients died within 12 months, of whom, two (3.7%) within 6 months (Table 2). The most frequent reasons for dose reduction were gastrointestinal side effects (7 (13.0%)), in particular diarrhea (7 (13.0%)) and liver enzyme derangement (6 (11.1%)) (Table 3). Similarly, gastrointestinal intolerance (11 (20.4%)) was the main reason for treatment suspension (Table 3). The mean (SD) time to dose reduction was 4.2 (2.7) months, and the mean (SD) time to suspension was 6.8 (3.3) months.

The PFTs showed no significant changes in the FVC (*p* = 0.469) and the FEV1 (*p* = 0.949) between the baseline and 6 months in the subgroup of patients who had been treated with Nintedanib for at least 4 months. In contrast, the DLCO measurements showed a significant reduction over the same period (*p* = 0.041). Similarly, between the baseline and 12 months in the subgroup of subjects treated with the study drug for at least 9 months (Figure 1), the changes in the FVC and FEV1 were not statistically significant (*p* = 0.396 and *p* = 0.055, respectively), while the DLCO was significantly reduced (*p* = 0.048). No significant results were obtained in the male-only subgroup analysis. The PFTs results are reported in Table 4 and Table A1 (see Appendix A).

Analyzed separately (Table 5), the subgroup of patients who completed full-dose treatment and those who reduced dosage showed no significant differences in the FVC and DLCO measurements between the baseline and 12 months. On the other hand, those who discontinued treatment showed a statistically significant reduction in the FVC measurements (*p* = 0.035) but not in the DLCO measurements.

The RR of progression at 12 months in the population of subjects who discontinued treatment, compared to those who continued treatment at a full or reduced dose, was 1.28 (95% CI: 0.572–2.863; *p* = 0.548).

The overall survival was 96.23% (95% CI: 0.857–0.990) at six months and 86.79% (95% CI: 0.743–0.935) at twelve months (Figure 2).

## 3. Discussion

### 3.1. Main Results

The aim of this retrospective study was to assess the effects of reducing or discontinuing Nintedanib (NTD) in treating 56 patients with Idiopathic Pulmonary Fibrosis (IPF). Segregating the cohort into those strictly adhering to the prescribed pharmaceutical dosage and those reducing dosage, there were no significant differences in the Forced Vital Capacity (FVC) and Diffusing Capacity of the Lungs for Carbon Monoxide (DLCO) measurements between the baseline and the 12-month mark. However, a notable observation emerged: the individuals who ceased treatment exhibited a statistically significant decline in the FVC measurements (*p* = 0.035) compared to their baseline. The assessment of the relative risk (RR) of progression after a 12-month period, comparing individuals who ceased treatment with those who maintained either full or reduced doses, resulted in a value of 1.28 (95% CI: 0.572–2.863; *p* = 0.548). Despite indicating a numerical advantage, this outcome lacks sufficient statistical validation.

### 3.2. Comparison with the Relevant Literature

Concerning medication adherence, this study revealed that gastrointestinal side effects, primarily diarrhea (63.6%), emerged as the primary reason for treatment suspension, as indicated in the Vincent Cottin et al. study [20]. Another significant adverse effect observed was hepatic enzyme derangement (54.5%). Elevations in liver enzymes and bilirubin, commonly observed within the initial three months of treatment, were usually reversible through dose interruption or reduction [21].

Comparative analysis with the existing scientific literature, particularly reviews on IPF and NTD therapy, reveals both confirmations and disparities [25,26,27,28,29,30,31,32,33]. The reduction in the DLCO aligns with antecedent studies, suggesting a potentially negative impact of this treatment on gas exchange. The TOMORROW trial evidenced a reduced annual decline in the FVC compared to a placebo [25,26,27]. Favorable outcomes similar to those in the TOMORROW trial were demonstrated in the INPULSIS trials, extending the time before the initial occurrence of acute exacerbation [25]. The INPULSIS-ON investigation further supported these findings, providing an enduring evaluation of the safety parameters and toxicity profiles associated with long-term NTD use [27].

Conversely, the lack of significant changes in the FVC contrasts with specific findings in the literature. The combined analysis of four placebo-controlled studies showed that Nintedanib reduced the decline rate in the FVC by approximately 50% within 52 weeks among individuals affected by various forms of pulmonary fibrosis [33]. The absence of significant changes in the FVC observed in our study may be attributed to various physiological and pathological variables. The heterogeneity of underlying lung pathology, such as Idiopathic Pulmonary Fibrosis (IPF), might influence treatment response. Additionally, genetic and metabolic differences among patients could affect drug absorption, metabolism, or response.

Concerning a real-life situation, Hirashawa et al. examined 86 IPF patients treated with NTD; in this group, 50 patients experienced Nintedanib-related diarrhea. The drug was continuously administered in 26 patients, 11 with full-dose medication and 15 with a reduction; in this case, in a one-year follow-up the therapy reduction had no impact on the FVC decline, as shown in our study. The authors observed that when diarrhea occurs within a year after using NTD, the dose reduction may be acceptable without affecting pulmonary function. Moreover, treatment with multiple antidiarrheals may be a practical option to maintain the use of NTD therapy compared with monotherapy and no therapy. In their study, the researchers used Clostridium butyricum as the probiotic bacterium most commonly used (in 28 of their 46 patients) as an antidiarrheal agent. In conclusion, it is important to underline that the use of antidiarrheal therapy, even with the association of multiple probiotic drugs, is a practical way to prolong the tolerability of NTD [34]. In another study, Fletcher et al. conducted a multi-center cohort review of 154 patients treated with NTD; in their study populations, the authors observed that 67.5% of patients presented diarrhoea and 7.2% discontinued as a result of diarrhoea; compared to our real-life study, the number of patients who presented diarrhea was higher, but the number of subjects who stopped treatment due to this symptom was lower. This data confirms what was suggested by Hirashawa et al. regarding the fact that in clinical practice it may be useful to manage diarrhea with symptomatic drugs to continue treatment with NTD [35]. Galli et al. conducted a real-life study on the tolerance of the two antifibrotic drugs Pirfenidone and NTD on 186 subjects. In patients treated with NTD, overall, 21% required dose reduction and 26.3% required discontinuation. The results on tolerance are substantially comparable to our study [36]. Even in the study of Hughes et al., diarrhoea was the main adverse event of NTD, usually mild to moderate but manageable. The authors make some interesting recommendations to manage it; in particular, they underline the importance of including rehydration and modulating the antidiarrheal medication. In addition, regarding the other most important gastrointestinal symptom, nausea, the researchers supported the use of omeprazole as a good therapy that can be given to patients taking NTD. Furthermore, they underline that, in the event that nausea persists despite dose adjustments, interventions such as an antiemetic can be introduced to provide symptomatic relief in order to continue treatment [37]. Even in this study the percentage of adverse events and the strategies to resolve them were similar to ours.

To our current knowledge, there exists only one documented study discussing the progression of the disease following the suspension or reduction of therapy with NTD. In a case report by Satoshi Okamori et al. [28], two cases of IPF showed a deterioration in respiratory conditions three weeks after discontinuation of NTD. These deteriorations were consistent with accelerated disease progression due to the absence of other potential causes exacerbating the respiratory condition.

Interestingly, studies on tyrosine kinase inhibitor (TKI) therapy for various cancers reported disease flare-ups after TKI discontinuation. Other studies on renal carcinoma demonstrated disease resurgence due to rapid angiogenesis after discontinuation of the TKIs targeting the VEGF and PDGF receptors. Multiple protein tyrosine kinases have been implicated in fibrosis development and progression.

Reactivation of various tyrosine kinase receptors, including the VEGF and PDGF receptors, following TKI discontinuation, may lead to accelerated disease progression [29,30,31]. However, the precise mechanism underlying this effect remains to be elucidated.

### 3.3. Theoretical Implications

The study results carry theoretical implications for understanding the specific impact of Nintedanib on pulmonary function in patients with Idiopathic Pulmonary Fibrosis (IPF). Practical applications encompass the importance of meticulous monitoring for side effects, particularly gastrointestinal manifestations, and of considering personalized treatment plans to optimize adherence and to reduce adverse effects.

### 3.4. Limitations and Strengths

The limitations of this study include its retrospective design across two centers and a reasonably sized sample, according to the low prevalence of the disease (3–9 cases per 100,000 individuals in Europe and North America) [1,2,3,4]. Notable missing data, even in repeated FVC measurements, could introduce biases. Among the strengths, a robust methodological approach was employed, accurately evaluating the reasons for treatment interruption or reduction and offering a comprehensive view of disease progression concerning changes in the therapeutic regimen. A detailed analysis of complications or adverse effects following NTD cessation or reduction can contribute to a better understanding of the associated risks and benefits with these therapeutic decisions.

### 3.5. Recommendations for Future Research

This study provides valuable insights into the clinical trajectory of IPF patients undergoing NTD treatment after reduction or cessation. Despite an overall high survival rate, concerns arise regarding the drug’s impact on the DLCO and potential consequences upon discontinuation. Ongoing research is imperative to comprehend the intricate relationship between NTD, pulmonary function, and long-term prognostic outcomes within this patient cohort. Conducting further longitudinal clinical studies is crucial to fully grasp the variability of NTD response and to identify predictive markers guiding personalized management for IPF patients or other lung pathologies treated with this drug.

## 4. Materials and Methods

### 4.1. Study Population and Design of the Study

In this retrospective analysis, eligible participants included all adults (aged 18 years or older) diagnosed with IPF and who started NTD treatment between 2015 and 2022 at either study center (the Department of Pulmonology, University Hospital of Cattinara, Trieste, Italy; and the Pulmonology Unit, General Hospital “Santa Maria degli Angeli”, Pordenone, Italy). We excluded subjects with insufficient clinical information (i.e., age at diagnosis and smoking habits), with no available data regarding blood tests or diagnostic imaging results, with active cancer, and with connective tissue diseases.

The diagnosis of IPF was made according to the 2018 ATS/ERS/JRS/ALAT Clinical Practice Guideline [2]. At baseline, all patients started NTD; as per EMA/AIFA authorization, 150 mg twice daily was considered a full dose and 100 mg twice daily was considered a reduced dose. All patients underwent specialist discussion prior to the modification of the treatment protocol. The assessment of ILD in our centers was based on HRCT and was conducted by radiologists possessing over a decade of expertise in lung imaging.

### 4.2. Data Collection

Clinical and functional data were gathered retrospectively at baseline, as well as at the 6-month and 12-month marks following the initiation of NTD. The following clinical features were collected: sex, age at baseline, age at disease onset, smoking history, comorbidities (GERD, COPD, and cancer), concomitant medications (OCS, hydroxychloroquine, azathioprine, methotrexate, mycophenolate mofetil, rituximab, tocilizumab, and PPI), NTD therapeutic dose, tolerability and adverse events, changes in treatment protocol and time to change, mortality, and time to death.

### 4.3. Pulmonary Function Tests

Lung function measurements were gained in adherence to ATS/ERS standards [32]. These parameters included the Forced Expiratory Volume in one second (FEV1), Forced Vital Capacity (FVC), and Diffusing Capacity of the Lungs for Carbon Monoxide (DLCO). The FVC, FEV1, and DLCO were described as a percentage of the predicted values for the patient’s age, sex, and height. Abnormalities in the PFTs were defined as predicted values of the FVC < 80% and the DLCO < 70%.

### 4.4. Statistical Analysis

Continuous variables were summarized using either the mean (standard deviation) or median (interquartile range) as deemed appropriate, while categorical variables were expressed as percentages. The normality of continuous variables was assessed using the Shapiro–Wilk test. Regarding the spirometry measurements, comparisons between the values at different time points were conducted using the paired *t*-test or the non-parametric Wilcoxon signed-rank test when assumptions for the parametric test were not met.

When comparing measurements at baseline to those at 6 months, the analysis only included individuals who had been on NTD for a minimum of 4 months. Similarly, when comparing measurements at baseline and at 12 months, only individuals who had been on NTD for at least 9 months were considered. The same analysis was also carried out in the male cohort. Furthermore, the PFTs measurements (the FVC and DLCO) were compared at baseline and 12 months within the subgroups of patients who either maintained full-dose treatment, underwent a dose reduction, or discontinued treatment.

The relative risk (RR) of progression (defined as a 12-month decline in FVC > 5% or death) [9] was calculated in the cohort of patients who discontinued treatment compared to those who completed the 12-month follow-up, either at a full or reduced dose.

A survival analysis was conducted, taking the initiation of NTD as the baseline and considering a one-year observation period. The endpoint was defined as death from any cause, and the survival function was estimated using the Kaplan–Meier method.

The statistical significance was set at 0.05. All the analyses were conducted using Jasp software version 18.1.

## 5. Conclusions

Nintedanib has demonstrated efficacy in mitigating a decline in the forced vital capacity (FVC) among the subgroups of patients adhering to the full or reduced dosing regimen. Notably, the patients who discontinued treatment showed a significant decline in the FVC, outlining the concerning impact of therapy suspension on pulmonary function and the importance of maintaining treatment continuity to potentially mitigate functional deterioration. The observed tolerability and efficacy of Nintedanib, together with the documented adverse reactions, are consistent with data from the existing literature.

In conclusion, our findings highlight the intricate balance between the efficacy and tolerability of Nintedanib, highlighting the need for further research to refine treatment protocols and optimize patient outcomes.

## Figures and Tables

**Figure 1 pharmaceuticals-17-00119-f001:**
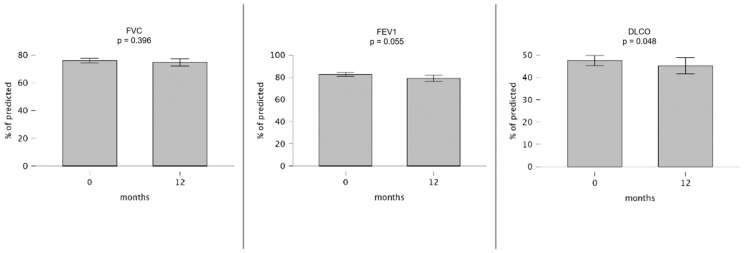
Comparison of the measurements of the pulmonary function tests after 12 months. Bar-plots representing levels of FVC, FEV1, and DLCO at baseline and after 12 months.

**Figure 2 pharmaceuticals-17-00119-f002:**
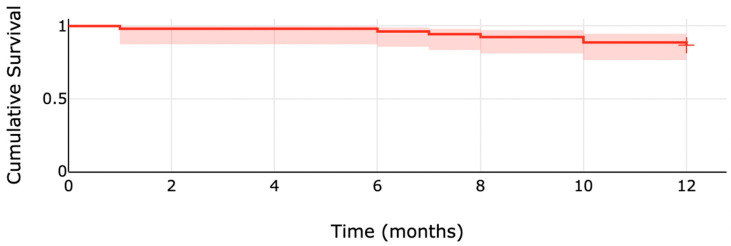
Kaplan–Meier estimates of 12 months survival probability.

**Table 1 pharmaceuticals-17-00119-t001:** Baseline characteristics of the study population.

	*n* = 54
Age at baseline (years), median (IQR)	75.0 (69.0–79.0)
Age of onset (years), median (IQR)	73.5 (68.0–77.0)
Gender	
Males	38 (70.4)
Females	16 (29.6)
Ever-smoker	31 (57.4)
Previous coexisting disease	
GERD	16 (29.6)
COPD	5 (9.3)
Cancer	7 (13.0)
Concomitant medication	
OCS	17 (31.5)
Hydroxychloroquine	0 (0.0)
Azathioprine	1 (1.9)
Methotrexate	0 (0.0)
Mycophenolate mofetil	2 (3.7)
Rituximab	0 (0.0)
Tocilizumab	0 (0.0)
PPI	35 (64.8)
HRCT pattern	
UIP	38 (70.4)
NSIP	9 (16.7)

Data are presented as No. (%), unless otherwise stated. IQR: interquartile range; GERD: gastroesophageal reflux disease; COPD: chronic obstructive pulmonary disease; OCS: oral corticosteroids; PPI: proton pump inhibitors; HRCT: high-resolution computed tomography; UIP: usual interstitial pneumonia; and NSIP: non-specific interstitial pneumonia.

**Table 2 pharmaceuticals-17-00119-t002:** Completion of therapy and mortality at 12 months.

	*n* = 54
Completion of therapy ^#^	
Full dose	20 (37.0)
Reduced dose	11 (20.4)
Suspended	15 (27.8)
Mortality	
6 months	2 (3.7)
12 months	7 (13.0)

Data are presented as No. (%). ^#^: 6 missing data.

**Table 3 pharmaceuticals-17-00119-t003:** Causes of dose reduction and treatment suspension at 12 months.

Dose Reduction	*n* = 11
Gastrointestinal intolerance	7 (13.0)
Diarrhea	7 (13.0)
Nausea/vomiting	1 (1.9)
Abdominal pain	1 (1.9)
Weight loss	1 (1.9)
Allergic reaction	0 (0.0)
Skin ulcer	0 (0.0)
Cough	0 (0.0)
Upper respiratory tract infection	0 (0.0)
Fatigue	0 (0.0)
Liver enzyme derangement	6 (11.1)
Treatment suspension	*n* = 15
Gastrointestinal intolerance	11 (20.4)
Diarrhea	8 (14.8)
Nausea/vomiting	2 (3.7)
Abdominal pain	1 (1.9)
Weight loss	2 (3.7)
Allergic reaction	0 (0.0)
Skin ulcer	0 (0.0)
Cough	0 (0.0)
Upper respiratory tract infection	0 (0.0)
Fatigue	0 (0.0)

Data are presented as No. (%).

**Table 4 pharmaceuticals-17-00119-t004:** Comparison of the measurements of the pulmonary function tests after 6 and 12 months.

Baseline	*n*	6 Months	*n*	*p*	Baseline	*n*	12 Months	*n*	*p*
FVC									
74.0 (62.5–82.5)	39	73.0 (64.0–84.5)	31	0.469	75.0 (64.0–82.5)	35	76.0 (58.0–84.5)	23	0.396
FEV1									
81.0 (69.0–94.5)	35	81.0 (67.5–90.5)	27	0.949	81.0 (70.0–97.0)	31	75.0 (69.5–86.8)	18	0.055
DLCO									
43.0 (34.0–60.0)	37	38.0 (31.0–48.8)	26	0.041	44.0 (34.0–60.8)	34	42.5 (32.5–57.8)	22	0.048

Data are presented as median (IQR). FVC: forced vital capacity; FEV1: forced expiratory volume in 1 s; and DLCO: diffusing capacity of the lungs for carbon monoxide. When comparing the measurements at baseline to those at 6 months, the analysis only included individuals who had been on NTD for a minimum of 4 months; when comparing measurements at baseline and at 12 months, only individuals who had been on NTD for at least 9 months were considered.

**Table 5 pharmaceuticals-17-00119-t005:** Comparison of the measurements of the pulmonary function tests at 12 months based on the completion of the therapy.

	Baseline	*n*	12 Months	*n*	*p*
Full dose					
FVC	74.0 (66.0–81.0)	17	66.0 (61.0–82.0)	11	0.975
DLCO	44.0 (35.0–58.5)	18	44.0 (29.8–53.8)	12	0.098
Reduced dose					
FVC	81.0 (68.0–87.5)	10	81.0 (77.0–89.0)	7	0.880
DLCO	46.0 (38.0–58.0)	8	38.5 (37.3–50.3)	6	0.084
Suspended					
FVC	70.0 (59.0–82.0)	13	55.0 (52.8–66.3)	8	0.035
DLCO	43.0 (31.8–50.5)	12	37.5 (31.0–55.3)	6	0.350

Data are presented as median (IQR). FVC: forced vital capacity; and DLCO: diffusing capacity of the lungs for carbon monoxide.

## Data Availability

Deidentified participant data will be made available upon motivated request to the Corresponding Author.

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
