# Peer review of "Functional Progression after Dose Suspension or Discontinuation of Nintedanib in Idiopathic Pulmonary Fibrosis: A Real-Life Multicentre Study"

_pharmaceuticals, 2024, doi:10.3390/ph17010119_

Round 1
Reviewer 1 Report
Comments and Suggestions for Authors
pharmaceuticals-2779544
General comments
This retrospective study is to assess the effects of reducing or discontinuing nintedanib in patients with IPF. Many studies on the safety and tolerability of approved antifibrotics including nintedanib have revealed that intolerance due to adverse events is common, resulting in dose reduction or even discontinuation. As the authors mentioned this study would provide valuable insights into the clinical trajectory of IPF patients undergoing nintedanib treatment after reduction or cessation. I would like to make sure that there were no patients who discontinued nintedanib due to IPF progression or acute exacerbation of IPF in this study population. These may affect trajectory of FVC in the treatment suspension group.
Minor comments
Abstract
Line 32-36:
DLCO at both 6 (38.0±17.8 vs. 43.0±26.0;・・・, FVC・・・
Are these figures shown as % predicted? Please add the units.
1. Introduction
Line 86
Nintedanib→NTD
2. Result
Line 92,93
The most common concomitant medications were PPI and OCS・・
According to the current guidelines on therapeutic strategies for IPF (Am J Respir Crit Care Med. 2015;192:e3-19), the recommendation against the use of prednisone is strong. There are 17 patients (31.5%) using OCS. The reason for the use of OCS should be described.
Line 94-95
while 9 (16.7%) presented a NSIP pattern.
How to diagnose these patients with a NSIP pattern on HRCT? Number of subjects underwent lung biopsy (VATS or TBLC) should be added in the Table1.
Line 98
6 (11.1%) subjects did not report any data.
These subjects may represent lost-to follow up. The authors should mention the reason for no data.
Table 3
Line 106
‘Treatment suspension’ should be indexed as same as ‘Dose reduction’.
Table 4
Line 115
Table 4 may get confusing. After reading the ‘4.4. Statistical Analysis’ describing that when comparing measurements at baseline to those at 6 months, the analysis included only individuals who had been on NTD for a minimum of 4 months, the reason why the baseline number of subjects are different can be recognized. I recommend these comments as the footnote of Table 4.
3. Discussion
Line 162
The combined analysis of four placebo-controlled studies showed that nintedanib reduced the decline rate in FVC by approximately 50% within 52 weeks ・・・
The authors should cite the paper.
Author Response
General comments
This retrospective study is to assess the effects of reducing or discontinuing nintedanib in patients with IPF. Many studies on the safety and tolerability of approved antifibrotics including nintedanib have revealed that intolerance due to adverse events is common, resulting in dose reduction or even discontinuation. As the authors mentioned this study would provide valuable insights into the clinical trajectory of IPF patients undergoing nintedanib treatment after reduction or cessation. I would like to make sure that there were no patients who discontinued nintedanib due to IPF progression or acute exacerbation of IPF in this study population. These may affect trajectory of FVC in the treatment suspension group.
Dear Reviewer,
Thank you very much for reviewing our manuscript and we really appreciate your helpful comments and valuable suggestions, which not only helped us with the improvement of our manuscript but also will facilitate our future research. Based on these comments and suggestions, we carefully revised and improved our manuscript. We are now sending the new manuscript for your reconsideration to publish in Pharmaceuticals and we hope the revisions will meet your approval. Our point-by-point responses to the comments and suggestions are as follows, and please see the new manuscript for details of revisions. The newly written parts for revision are marked with red color.
Thank you very much for considering our research appropriate and informative. We confirm that there were no patients who discontinued nintedanib due to IPF progression or acute exacerbation of IPF in this study population.
Minor comments
1) Abstract
Line 32-36:
DLCO at both 6 (38.0±17.8 vs. 43.0±26.0;・・・, FVC・・
Are these figures shown as % predicted? Please add the units.
R: Thank you very much. We add the units as suggested.
2) 1. Introduction
Line 86
Nintedanib→NTD
R: Thank you. We correct the error.
3) 2. Result
Line 92,93
The most common concomitant medications were PPI and OCS・・
According to the current guidelines on therapeutic strategies for IPF (Am J Respir Crit Care Med. 2015;192:e3-19), the recommendation against the use of prednisone is strong. There are 17 patients (31.5%) using OCS. The reason for the use of OCS should be described.
R: Thank you very much for this observation, which give us the possibility to specify that OCS treatment was present at baseline at the first visit at our center. In many cases, the treatment was introduced by other clinicians before the definitive diagnosis, which occurred at our center also thanks to the re-evaluation and discussion of the cases in the multi-disciplinary team. Once the diagnosis was made, the treatment was suspended. We modify table 1 and the text as reported: “Baseline characteristics, collected at the time of the first visit in our center and before any treatments’ modification, are reported in Table 1.”
4) Line 94-95
while 9 (16.7%) presented a NSIP pattern.
How to diagnose these patients with a NSIP pattern on HRCT? Number of subjects underwent lung biopsy (VATS or TBLC) should be added in the Table1.
R: We would like to thank the reviewer for this appropriate comment and in accordance with the reviewer's suggestion, we have added the information in the text, in particular “All cases considered were discussed in multidisciplinary meetings, 10 cases (18%) in which the differential diagnosis was not certain underwent lung biopsy which confirmed the diagnosis of IPF”
5) Line 98
6 (11.1%) subjects did not report any data.
These subjects may represent lost-to follow up. The authors should mention the reason for no data.
R: We would like to thank the reviewer for this comment. Some patients came from smaller centers in the area or in neighboring regions, who trusted us to carry out the diagnosis and start therapy and were then re-entrusted to these centres, unfortunately, we lost their follow-up. In addition, some patients did not perform functionally test as scheduled by our follow up so we decided to exclude this data in our collection. We specify this in the text, in particular “All included patients started therapy at baseline with a dosage of 300 mg per day, 12 months later 20 (37.0%) were still taking Nintedanib at full dose, 11 (20.4%) had reduced it to 200 mg daily and 15 (27.8%) had stopped treatment. 6 (11.1%) subjects did not report any data; of these, some patients were not considered as the functional follow-up did not take place as we scheduled, so it was decided not to include them. Some of the patients, coming from centers far from ours, preferred to continue treatment near their home.”
6) Table 3
Line 106
‘Treatment suspension’ should be indexed as same as ‘Dose reduction’.
R: Thank you very much, we modify the table as requested
7) Table 4
Line 115
Table 4 may get confusing. After reading the ‘4.4. Statistical Analysis’ describing that when comparing measurements at baseline to those at 6 months, the analysis included only individuals who had been on NTD for a minimum of 4 months, the reason why the baseline number of subjects are different can be recognized. I recommend these comments as the footnote of Table 4.
R: Thank you for your correction, in agreement with your comment we specify the methods of data collection also in table 4. In particular “Data are presented as median (IQR). FVC: forced vital capacity; FEV1: forced expiratory volume in 1 second; DLCO: diffusing capacity of the lungs for carbon monoxide. When comparing measurements at baseline to those at 6 months, the analysis included only individuals who had been on NTD for a minimum of 4 months; when comparing measurements at baseline and at 12 months, only individuals who had been on NTD for at least 9 months were considered.”
8) 3. Discussion
Line 162
The combined analysis of four placebo-controlled studies showed that nintedanib reduced the decline rate in FVC by approximately 50% within 52 weeks ・・・The authors should cite the paper.
R: Thanks for this suggestion. As required, we added the paper in the text.
Reviewer 2 Report
Comments and Suggestions for Authors
The study is very important, and the results are needed and expected.
However, the study has several quite important aspects to notice:
- NSIP radiological pattern is not a pattern for IPF – I suspect that the study group was not limited to IPF patients only. So in my opinion the title and parts of the text regarding the study group need to be changed because the study group did not include only patients with IPF
- in connection: it was a high % on OCS treatment (31.5%) – what is an explanation? It is not a recommended therapy for IPF patients
- GERD was occurred in 29.6% - it is a low percentage, but PPI was used in 64.8% - what is the comment?
- I also have doubts whether the title of the manuscript should include the term "multicenter", since the study concerns two centers?
- whether patients with a radiological pattern of NSIP had features of fibrotic NSIP?
- what were the inclusion and exclusion criteria for the study? – it should be better defined
- a small group of patients - quite a long study period, two centers and only such a number of patients? The small number of patients is particularly noteworthy in Table 6 when the results of lung function tests are assessed over time, taking into account the dose of nintedanib therapy - "n" is then several - the conclusions are very limited
- when presenting the results as a median, the value of the 1st and 3rd quartiles is usually placed in brackets - in the presented work only one unspecified value is. In the "statistical methods" section, writing about the median (interquartile range), but the results do not provide a range, only one value - this requires correction
- when side effects are reported, it will be more useful to state what percentage they represented in the entire study group, not only among patients who had side effects - table 3
- table 4 shows not only spirometric values but also DLCO, so it could be described rather as: "Comparison of lung function test results after 6 and 12 months" (the same applies to Figure 1 and table 6)
- when it comes to definitions, it might be worth specifying how long during these 12 months patients were treated with nintedanib at a reduced dose - it is worth specifying this even if it was the entire duration of treatment. If there was a complete cessation of treatment, how long after taking nintedanib did it occur?
- the discussion did not refer to the results of real-word studies regarding the use of nintedanib at a reduced dose / discontinuation of treatment - and these comparisons will be more valuable than referring to clinical trials where the groups are disproportionately larger than in the presented study
- It would be appropriate to refer to the results of the work of: Hirasawa Y et al. Tolerability of nintedanib-related diarrhea in patients with IPF. Pulmon Pharm Ther 2020; 62 , where the authors also showed similar effectiveness in slowing down the progression of the disease in patients using the full and reduced dose of nintedanib - and this is probably the most important conclusion that it is better to keep a patient with pulmonary fibrosis on therapy, even at a reduced dose, than to discontinue the treatment
Author Response
The study is very important, and the results are needed and expected.
Dear Reviewer,
first of all, we would like to thank you for giving a chance to revision of the our manuscript. Taking the suggestions and the constructive criticism of the valuable referee into account, we have tried to do our best in the revision of article. We would also like to thank the referee for the guidance and the contributions to article. The newly written parts for revision are marked with red color.
However, the study has several quite important aspects to notice:
- NSIP radiological pattern is not a pattern for IPF – I suspect that the study group was not limited to IPF patients only. So in my opinion the title and parts of the text regarding the study group need to be changed because the study group did not include only patients with IPF.
R: We would like to thank the reviewer for this appropriate comment and in accordance with the reviewer's suggestion, we have added the information in the text, in particular, “All cases considered were discussed in multidisciplinary meetings, 10 cases (18%) in which the differential diagnosis was not certain underwent lung biopsy which confirmed the diagnosis of IPF”.
- in connection: it was a high % on OCS treatment (31.5%) – what is an explanation? It is not a recommended therapy for IPF patients
R: Thank you very much for this observation, which give us the possibility to specify that OCS treatment was present at baseline at the first visit at our center. In many cases, the treatment was introduced by other clinicians before the definitive diagnosis, which occurred at our center also thanks to the re-evaluation and discussion of the cases in the multi-disciplinary team. Once the diagnosis was made, the treatment was suspended. We modify table 1 and the text as reported: “Baseline characteristics, collected at the time of the first visit in our center and before any treatments’ modification, are reported in Table 1.”
- GERD was occurred in 29.6% - it is a low percentage, but PPI was used in 64.8%. What is the comment?
R: We agree with the reviewer regarding the high percentage of PPI treatment in our cohort of patients, as previously reported this is the baseline situation of the treatment collected with the anamnesis at the first visit in our center.
- I also have doubts whether the title of the manuscript should include the term "multicenter", since the study concerns two centers?
R: Thank you very much we correct the title, as reported: "Functional progression after dose suspension or discontinuation of Nintedanib in idiopathic pulmonary fibrosis: a two centers real-life study".
- whether patients with a radiological pattern of NSIP had features of fibrotic NSIP?
R: Thank you for your comment. As previously reported all cases were revied in a multidisciplinary team discussion with radiologists and clinicians expert in the fields. In particular, we added in the text “All cases considered were discussed in multidisciplinary meetings, 10 cases (18%) in which the differential diagnosis was not certain underwent lung biopsy which confirmed the diagnosis of IPF”.
- what were the inclusion and exclusion criteria for the study? It should be better defined, a small group of patients, quite a long study period, two centers and only such a number of patients? The small number of patients is particularly noteworthy in Table 6 when the results of lung function tests are assessed over time, taking into account the dose of nintedanib therapy - "n" is then several - the conclusions are very limited
R: Thank you very much for your comment. We improve “material and methods” adding inclusion and exclusion criteria. Regarding the study populations, we specify that the number of patients is low, according to the low prevalce of the disease and we added this data both in the introduction chapter and in the discussion chapter. In addition, patients with IPF characteristics who did not satisfied AIFA criteria for Nintedanib prescription were excluded.
- when presenting the results as a median, the value of the 1st and 3rd quartiles is usually placed in brackets - in the presented work only one unspecified value is. In the "statistical methods" section, writing about the median (interquartile range), but the results do not provide a range, only one value - this requires correction
R: Thank you for your comment, the IQR has been expressed as suggested.
- when side effects are reported, it will be more useful to state what percentage they represented in the entire study group, not only among patients who had side effects - table 3
R: Thank you for your suggestion, side effects are now reported as a percentage of the entire study population.
- table 4 shows not only spirometric values but also DLCO, so it could be described rather as: "Comparison of lung function test results after 6 and 12 months" (the same applies to Figure 1 and table 6)
R: Thank you for the observation, “spirometric measurements” has been changed to “pulmonary function tests measurements” as appropriate.
- when it comes to definitions, it might be worth specifying how long during these 12 months patients were treated with nintedanib at a reduced dose - it is worth specifying this even if it was the entire duration of treatment. If there was a complete cessation of treatment, how long after taking nintedanib did it occur?
R: Thank you for the comment. Mean time to dose reduction and mean to treatment suspension have now been included in the main text. No patient started treatment at reduced dose, the introduction of the results’ second paragraph has been rearranged to better clarify this concept to “All included patients started therapy at baseline with a dosage of 300 mg per day […]”.
- the discussion did not refer to the results of real-word studies regarding the use of nintedanib at a reduced dose / discontinuation of treatment - and these comparisons will be more valuable than referring to clinical trials where the groups are disproportionately larger than in the presented study. It would be appropriate to refer to the results of the work of: Hirasawa Y et al. Tolerability of nintedanib-related diarrhea in patients with IPF. Pulmon Pharm Ther 2020; 62 , where the authors also showed similar effectiveness in slowing down the progression of the disease in patients using the full and reduced dose of nintedanib - and this is probably the most important conclusion that it is better to keep a patient with pulmonary fibrosis on therapy, even at a reduced dose, than to discontinue the treatment
R: Thank you very much, your suggestions will be of great help to us to improve the quality of our papers. We added some studies about real-life experience and we also inserted the work you suggested, stressing the fact that it is important to keep a patient with pulmonary fibrosis on therapy, even at a reduced dose, than to discontinue the treatment. In particular “Concerning real life experiences, Hirashawa et al. examined 86 IPF patients treated with Nintedanib, in this group 50 patients experienced Nintedanib-related diarrhea. The drug was continuously administered in 26 patients, 11 with full dose medication and 15 with reduction; in this case on one-year period follow up the therapy reduction has no impact on FVC decline, as shown in our study. Furthermore, the use of antidiarrheal therapy is a practical way to prolong the tolerability of Nintedanib [34]. Fletcher et al. conducted a multicentre cohort review on 154 patients treated with Nintedanib, in their study populations the authors observed that 67.5% of patients presented diarrhoea and 7.2% discontinued as a result of diarrhoea; compared to our real life study, the number of patients who presented diarrhea was higher, but the number of subjects who stopped treatment due to this symptom was lower. This data confirms what was suggested by Hirashawa et al. regarding the fact that in clinical practice it may be useful to manage diarrhea with symptomatic drugs to continue treatment with Nintedanib [35]. Galli et al conducted a real life study on the tolerance of the two antifibrosant drugs Pirfenidone and Nintedanib on 186 subjects. In patients treated with Nintedanib, overall 21% required dose reduction and 26.3% required discontinuation. The results on tolerance are substantially comparable to our study [36]. “
Round 2
Reviewer 1 Report
Comments and Suggestions for Authors
According to the comments I made the authors have revised the manuscript well.
Reviewer 2 Report
Comments and Suggestions for Authors
I do not have more comments